# Purification and Characterization of Bot33: A Non-Toxic Peptide from the Venom of *Buthus occitanus tunetanus* Scorpion

**DOI:** 10.3390/molecules27217278

**Published:** 2022-10-26

**Authors:** Rym ElFessi, Oussema Khamessi, Najet Srairi-Abid, Jean-Marc Sabatier, Jan Tytgat, Steve Peigneur, Riadh Kharrat

**Affiliations:** 1Laboratoire des Venins et Biomolécules Thérapeutiques, Institut Pasteur de Tunis, Université de Tunis El Manar, 13 Place Pasteur BP74, Tunis 1002, Tunisia; 2Laboratoire Biomolécules, Venins et Applications Théranostiques (LR20IPT01), Institut Pasteur de Tunis, Tunis El Manar, 13 Place Pasteur BP74, Tunis 1002, Tunisia; 3Institut de Neurophysiopathologie (INP), Université Aix-Marseille, UMR 7051, 13005 Marseille, France; 4Toxicology and Pharmacology, Campus Gasthuisberg, University of Leuven (KU Leuven), 3000 Leuven, Belgium

**Keywords:** scorpion venom, potassium channel–Kv, voltage-dependent potassium channel

## Abstract

Scorpion venom is a rich source of promising therapeutic compounds, such as highly selective ion channel ligands with potent pharmacological effects. Bot33 is a new short polypeptide of 38 amino acid residues with six cysteines purified from the venom of the *Buthus occitanus tunetanus* scorpion. Bot33 has revealed less than 40% identity with other known alpha-KTx families. This peptide displayed a neutral amino acid (Leucine), in the position equivalent to lysine 27, described as essential for the interaction with Kv channels. Bot33 did not show any toxicity following i.c.v. injection until 2 µg/kg mouse body weight. Due to its very low venom concentration (0.24%), Bot33 was chemically synthesized. Unexpectedly, this peptide has been subjected to a screening on ion channels expressed in *Xenopus laevis* oocytes, and it was found that Bot33 has no effect on seven Kv channel subtypes. Interestingly, an in silico molecular docking study shows that the Leu27 prevents the interaction of Bot33 with the Kv1.3 channel. All our results indicate that Bot33 may have a different mode of action from other scorpion toxins, which will be interesting to elucidate.

## 1. Introduction

Animal venoms are rich sources of active compounds with highly selective and potent pharmacological effects on potassium channels. These peptide compounds are considered promising tools for the development of potential new drugs against different channelopathies [1,2,3].

Indeed, animal toxins, initially studied to obtain effective anti-venoms, appear today as a tool of choice for developing selective blockers in order to study the functioning of the various channels.

The potassium channel toxins (KTx), isolated from scorpion venom, act by binding, with great affinity and more or less selectively, to different types of potassium channels which are involved in several physiological phenomena, including the regulation of membrane potential in excitable cells, such as neurons and muscles, as well as in non-excitable cells, such as pancreatic β-cells and T-cell lymphocyte proliferation [4,5].

Alpha-KTx toxins form the largest KTx family, and 31 subfamilies have been described [6]. They are short-chain peptides of 23 to 42 amino acid residues, cross-linked by three to four disulfide bridges [7]. Their secondary structure finds a common architectural motif formed by an alpha helix connected to two or three antiparallel beta sheets named the “Cysteine-stabilized alpha/beta motif” [8].

Each member of the alpha-KTx family exhibits against voltage-gated (Kv) potassium channels subtypes.

The *Buthus occitanus tunetanus* scorpion (Appendix A) represents one of the most dangerous species. Only a few peptides have been reported from its venom [9,10,11,12,13]. In our research study, we aimed to identify other peptides to better understand the effect and mode of action of this venom. 

In previous studies, we reported that the peptides Kbot55 and Kbot21 [6,14] purified from the venom of *Buthus occitanus tunetanus* target voltage-gated potassium channels with high affinity.

Here, we describe a new natural short peptide (4.3 KDa) from the venom of the scorpion *Buthus occitanus tunetanus* called Bot33.

It is the first scorpion venom-derived peptide presenting a Leu in position 27 instead of Lys found on all active Ktx peptides and considered a crucial structural element that constitutes, with an aromatic residue (Tyr/Phe), a functional dyad regarding binding properties for targeting Kv channels [15,16,17,18,19]. Its pharmacological characterization suggests that it belongs to a new scorpion toxin family.

## 2. Results

### 2.1. Purification of Bot33

The toxic BOTG50 fraction obtained from Sephadex G-50 chromatography of *Buthus occitanus tunetanus* scorpion venom was fractionated by HPLC on C18 reverse phase-HPLC (Figure 1A).

The fraction B7 containing peptides in the range of 2.1 to 4.3 KDa, according to SDS-PAGE, was collected, lyophilized, and loaded onto an analytical C18 reverse-Phase column (Beckman Coulter PN 235332, California, USA) and this fraction was eluted by a linear gradient (10 % to 35% for 50 min, then from 35% to 75% for 17 min) of acetonitrile. Fraction B7 was collected and injected into the same column with a steeper gradient (18 to 30% for 100 min) (Figure 1B). The chromatogram showed that B7 was composed of three fractions. The most representative fraction B’7 was collected and analyzed with a linear gradient (14% to 20% for 100 min) and allowed to purify the Bot33 peptide eluted at 48.42 min with 17% acetonitrile–TFA (Figure 1C).

### 2.2. Amino-acid Sequence of Bot33 and Sequence Analysis

The identification of the complete amino-acid sequence was carried out by the automatic Edman degradation of 2 nmoles of S-pyridylethylated peptides.

The Bot33 peptide is composed of 38 amino acid residues containing six cysteine residues colored yellow: TNGVPGKCTKPGGCSTYCRDTTGTMGLCKNSKCYCNKY. The experimental molecular weight of native Bot33 (4032.95 Da) obtained by MALDI ionization is very close to the average theoretical molecular weight calculated for the fully oxidized form of Bot33 (4035,62 Da) (Figure 1D).

### 2.3. Sequence Analysis

Bot33 has the sequence signature of the cysteine stabilized α/β (CSα/β) motif. Compared to short-chain toxins, Bot33 showed less than 40% identity with different alpha-KTx families [20,21], as assessed by multiple sequence alignment (MSA) (Figure 2). We have deposited the Bot33 sequence in the UniProt database “Section SPIN” under SPIN ID number S²PIN200024302.

### 2.4. Bot33 Does Not Exhibit Toxicity Following ICV Injection in Mice

When injected into black C57 mice, Bot33 did not show any toxicity up to 37 ng/20 g mice (2 µg/kg) body weight, as determined by i.c.v. injection. This novel peptide was at least ≈ 1500-fold less toxic than KTX (LD50 ≈ 24 ng/20 g mice) [22] (Appendix A).

### 2.5. Chemical Synthesis of Kbot3

The solid-phase chemical synthesis of the Kbot3 peptide was carried out on an Fmoc/t-butyl resin. The amount of target-protected peptide bound to the resin was 0.2 mmol, denoting an 80% assembly yield of peptide [23].

After lyophilization, the crude reduced peptide was solubilized under alkaline conditions while being exposed to air and with agitation for folding. The C18 reverse phase analytical HPLC profile before oxidation shows a relative homogeneity of the crude reduced peptide after treatment with TFA (Figure 3A). The HPLC profile shows different synthetic peptide forms (Figure 3B). After complete oxidation (for 72 h) in air, the chromatographic profile obtained by RP-C18 analytical reversed-phase HPLC of the purified oxidized Kbot3 peptide shows a homogeneity > 95% (Figure 3C).

Analysis of the purified peptide by MALDI-TOF mass spectrometry shows a single peak of experimental molecular mass (M + H)+ of 4039 Da which is compatible with the value (4041.62) derived from their primary structure (Figure 3D). Thus, the yield of chemical synthesis, which combines the yield of the final processing of TFA peptide, oxidative refolding, and purification, was 4%.

### 2.6. Effect of Synthetic Bot33 on Mice

The synthesized Bot33 peptide was assessed via intracerebroventricular injection for its effect on mice. Our results showed that it was not toxic up to 2 µg/kg mouse body weight (Appendix A).

### 2.7. Electrophysiological Experiments on Kv Channels Expressed in Xenopus Leavis Oocytes

Bot33 was screened against seven different Kv1.x channel subtypes using the two-electrode voltage-clamp method. At a concentration of 1 µM, Bot33 did not show any effect on the mammalian isoforms Kv1.1- Kv1.6. Moreover, no activity was seen on the insect Kv channel *Shaker* IR. As a control, ChTx was tested on Kv1.3 and *Shaker* IR. An amount of 1 µM of ChTx completely inhibited the current through these channels (Figure 4).

### 2.8. In Silico Study of Bot33 Interaction with Kv1.3 Channel

To check if channel Kv1.3, the target of the ChTx, is also targeted by Bot33, an in silico study was conducted.

### 2.9. Molecular Modeling of the Kv1.3 Potassium Channel, Bot33, and Its Toxin Mutations

The structures of Bot33 (Figure 5A) and Kv1.3 potassium channel receptor proteins were modeled using standard molecular modeling procedures [24]. Furthermore, we use in silico mutagenesis to obtain structural models of Bot33/M27 (Leu/Met27) (Figure 5B), and Bot33-K27_Y36 (Leu/Lys27 and Asn/Tyr36) by simply mutating the 27th (Leu to Lys or Met) and 36th (Asn to Tyr) amino acids, respectively (Figure 5C).

In silico mutagenesis and their 3D structures were predicted and evaluated by homology modeling, Ramachandran plot analysis (PROCHECK), ProsaII, and the Verify 3D (see Materials and Methods). Bot33 and the mutated peptides share the same Csαβ (Cysteine-stabilized αβ motif) fold formed by a helix (^13^GCSTYCRDT^21^**)** and two beta strands (^25^MGLCKNSKCYCN^34^) cross-linked by three disulfide bridges, C8–C28, C14–C33, and C18–C35 (Figure 5D).

### 2.10. Toxin–Channel Docking Study

The interaction of Bot33 with the Kv1.3 channel involves a beta-sheet residue. The analysis of the selected complex (toxin–channel) suggests that two regions able to interact with the interface of the channel were identified after filtering the clustering steps and molecular visualization. Alpha helix residues are not involved in the toxin–channel interaction. However, the residues of the beta-sheet are directly involved in the toxin–channel interaction. Several other amino acids of the toxin Bot33 are involved in the interaction with the Kv1.3 channel. We found that the majority of basic residues (Lys7, Lys10, Lys29, and Lys32) are implicated in the interaction. The interaction is stabilized only by a salt bridge between K^7^ and D^42^ with 2.1 Å. The residues T^9^, K^10^**,** T^16^, K^29^**,** and K^32^ interact with G^67^, G^285^, T^374^, S^155^, and T^44^ at 1.7 Å, 2.1 Å, 2.1 Å, 1.7 Å, and 1.7 Å, respectively (Figure 6A).

On the other hand, the docked complex (Bot33/M27–Kv1.3 channel) revealed the presence of seven interactions involved in extensive H-bonding and salt bridges. Amino acids T^1^, K^10^, K^29^**,** M^27^, N^30^, Y^34^, and Y^38^ of Bot33/M27 formed six hydrogen bonds with amino acids T^154^, T^374^, Y^396^, D^288^ H^180^, T^374^, and T^154^ of Kv1.3, respectively (Figure 6B). The distance between its interactions is of the order of 1.6 to 3.3 Å.

Further analysis showed that the protein complex between Bot33/K27_Y36 and the Kv1.3 channel was also stabilized by seven interactions (Figure 6C). Amino acids K^7^, K^10^**,** T^24^, K^27^**,** K^29^**,** N^30^, and Y^36^ of Bot33/K27_Y36 formed six hydrogen bonds with amino acids S^375^, D^264^, T^44^, Y^66^, D^288^, T^264^, and G^67^ of Kv1.3, respectively. The distance between its interactions is of the order of 1.7 to 2.7 Å. However, in Bot33/K27_Y36, only two stable interactions through salt bridges between K^10^ and K^29^ of the toxin and D^264^ and D^68^ of the channel, respectively, were observed. Figure 6C shows that the K^27^ of the toxin plays an important role in the interaction of the toxin–channel complex. The pore is physically blocked and shows that the K^27^ of the toxin interacts with the Y66 of the selectivity filter residues of the channel.

## 3. Discussion

Short scorpion toxins acting on potassium channels have been identified and classified into 32 subfamilies [25]. All these toxins are composed of fewer than 38 residues and exist in tiny amounts in the venom. Such peptides have generally been identified by bioassay-guided screening methods of the aqueous venom extract. In this work, we have used the same approach to identify a new non-toxic peptide which we named Bot33. This peptide is composed of 38 amino acids with six conserved cysteines.

Due to its very low concentration of venom, Bot33 was chemically synthesized by solid-phase peptide synthesis using the Fmoc/t-butyl strategy, and the purified oxidized Bot33 showed >95% homogeneity. Bot33 was screened on ion channels expressed in *Xenopus laevis* oocytes and it was found that Bot33 is not active on Kv1.1–Kv1.6 and *Shaker* IR channels. The electrophysiological results obtained showing the absence of inhibition of the Kv channel provide an explanation for the non-toxicity in vivo, observed in mice.

Compared to all members of KTx subfamilies, Bot33 showed a low amino acid sequence identity with these peptides, not exceeding 40% (Figure 2).

In a previous work, we identified a so-called scorpion toxin signature in the KTx families that are structurally and functionally important for binding to Kv channels. This scorpion toxin signature consists of six cysteines forming three disulfide bridges and two amino acids (Lys, Asn) in a four-residue-long motif (Lys–Cys4–Xaa–Asn, where Xaa is any amino acid), around the fourth cysteine [26,27]. This scorpion toxin signature is also conserved in Bot33 with the exception of the Lys in position 27, which is replaced by a Leu. The Lys in position 27 was reported to be essential, in alpha-KTx peptide structure, for their inhibitory activity on Kv channels. Dauplais et al. [19] reported that the absence of a functional dyad explains the lack of Kv channel inhibition in oocytes. Many K^+^ channel toxin inhibitors contain a critical positive charged residue (Lys or Arg) that protrudes into the channel pore and interacts with a Tyr residue situated in the selectivity filter of the K^+^ channel [27,28,29,30]. On the other hand, a hydrophobic residue (Tyr, Phe, Leu), contributes to the selectivity profile among different KV1.x channels [15,31,32]. Thus, this hydrophobic residue, with Lys 27 separated by a distance of 6.6 A, constitutes a functional dyad, reported to be essential for the high affinity toward Kv1.x channels.

The three-dimensional structure of Bot33 was modeled. Ribbon diagrams of Bot33 show an alpha/beta scaffold similar to those of other short scorpion toxins and containing a large hydrophobic and electropositive patch on the surface of the toxin. Our docking studies have shown that Bot33 interacts weakly with the Kv1.3 channel by essentially involving the basic residues located on the β sheet. During the in silico mutagenesis studies, initially, the substitution at position 27 of the Lys by the Met of Bot33, the Bot33/M27 complex, shows the involvement of seven interactions involved in an extended H bond and salt bridges. The Bot33/M27 toxin binds to the mouth of the pore of the Kv1.3 channel positioned to project the side chain of Lys29 directly into the pore against Y396 which acts as the essential lysine in the binding sites of other scorpion toxins, i.e., Lys-27 in ChTX. The aromatic ring of Y38 is positioned to pack against T154 in a similar position as Y38 to CHTX and M27 can package against D288; therefore, M27 allowed the change in the conformation of the toxin and increased the affinity of the toxin toward the channel, which confirms that the modification of a critical functional residue can markedly affect the activity of a peptide [33]. There are also residues N30 and Y34 which are involved in long-range electrostatic interactions. In a second stage of the mutagenesis studies, carried out by substitution of Leu27 after a Lys27 and N36 by Tyr 36 of Bot33, Bot33/K27-Y36 and the Kv1.3 channel form a stable interaction by a disulfide bond between K29 of the toxin and channel D68. The pore is physically blocked and shows that the K27 of the toxin interacts with the Y66 of the channel selectivity filter residues.

In fact, our work confirms the importance of the positively charged residue at position 27, and this has already been shown by other authors [27,28,29,30]. However, Naseem et al., [25] recently reported that the Cm28 peptide inhibited voltage-gated K^+^ channels KV1.2 and KV1.3 with Kd values of 0.96 and 1.3 nM, respectively, despite lacking this residue. In our work, we clearly showed that besides Lys27 other residues, such as hydrophilic residues T, K, and N, and hydrophobic residues M and Y, should be present in the toxin sequence to allow its interaction with the K^+^ channel. Therefore, it can be assumed that the Tyr36 and Lys27 residues play the role of the functional dyad in Bot33/K27-Tyr36. In future studies, it will be interesting to produce a Bot33 mutated to present this functional dyad and to verify its activity on Kv channels.

## 4. Materials and Methods

### 4.1. Scorpion Venom

The venom of the *Buthus occitanus tunetanus* scorpion, collected from Beni Khedach (Tunisia), was extracted by a veterinarian of the Pasteur Institute of Tunisia. The pooled venom was kept frozen at −20 °C in its crude form until use.

### 4.2. Purification of Bot33

Bot33 toxin was purified from the crude venom by two-step purification, consisting of gel filtration followed by reverse phase HPLC. The lyophilized crude venom was applied to a Sephadex G50 column equilibrated with 0.1 M ammonium acetate (pH 8.5). The eluted proteins were monitored at 280 nm. The biologically active fraction (BotG50) was then applied onto a C18 reverse-phase HPLC column (5 µm, 4.6 × 250 mm, Beckman Coulter) equilibrated by 0.1 % trifluoroacetic acid in water. The elution of the molecules was performed by a linear gradient of 0.1 % trifluoroacetic acid in acetonitrile (10 to 35% for 25 min), at a flow rate of 1ml/min. The detection was monitored at 215 nm. Fraction B7 containing the peptides in the range of 3.7–4.6 KDa, according to SDS-PAGE, was collected, lyophilized, and loaded under the same conditions and a larger gradient was applied for 50 min. The last fraction 7′7 was composed of three fractions when eluted by a linear gradient (14 to 20% for 100 min). (Figure 1). The most representative pure fraction 7′7c was selected for further characterization.

### 4.3. Amino Acid Sequence Determination

The reduction with dithiothreitol and the alkylation with 4-vinylpyridine of Bot33 and the determination of the sequence of the native and S-alkylated peptide was carried out as described by Srairi-Abid et al. [34]. Native Bot33 sequence analysis was performed with a reproducible 95% yield over 31 cycles of Edman degradation, and 1 nmol of S-alkylated protein was used to determine the sequence of Bot33 and to identify the positions of cysteine.

### 4.4. Mass Spectrometry

The peptide was analyzed on a voyager de RP MALDI-TOF mass spectrometer (Perspective Biosystems, Inc., Framingham, MA). The sample was dissolved in CH_3_CN/H_2_O (30/70) with 0.3% trifluoroacetic acid to obtain a concentration of 1–10 pmol/µL. The matrix was prepared as follows: alpha–cyanohydroxycinnamic acid was dissolved in 50% CH_3_CN in 0.3% trifluoroacetic acid/H_2_O to obtain a saturated solution at 10 µg/µL. A 0.5 µL of peptide solution was placed on the sample plate, and 0.5 µL aliquot of the matrix solution was added. This mixture was allowed to dry. Mass spectra were recorded in linear mode, externally calibrated with suitable standards, and analyzed by the GRAMS/386 software.

### 4.5. In Vivo Characterization of Bot33

The native and synthetic Bot33 peptides were diluted in 0.1% (*w/v*) BSA, and 5 µL of the solution containing increasing amounts of peptides were injected by the i.c.v. route into six C57/BL6 male mice of 20 ± 2 g, for each concentration [35].

### 4.6. Chemical Synthesis of Bot33

Fmoc [Nα-(9-fluorenyl) methoxycarbonyl]-L-amino acids and Fmoc resin for peptide synthesis, were obtained from Perkin-Elmer. Enzymes (trypsin and chymotrypsin) were obtained from Roche Molecular Biochemicals (Regensburg, Germany). Solvents of analytical grade were purchased from SDS (Solvents Documentation Synthesis, Address: quart Valdonne Postal code: 13124, Town/Department: Peypin, Bouches-du-Rhône). The synthetic Bot33 peptide was obtained by the solid phase technique [36], using a peptide synthesizer (model 433A; Applied Biosystems, CA, USA). The assembly of the peptide chains was carried out in stages on 0.25 mmol of Fmoc-amide resin (0.66 mmol of amino group/g) using 1 mmol of Fmoc amino acid derivatives. Side chain protecting groups for trifunctional residues were as follows: t-butyloxycarbonyl for Lys; t-butyl for Ser, Tyr, Thr, and Asp; trityl for Cys, Asn, and Gln; and pentamethyl chroman for Arg. The Fmoc amino acid derivatives were coupled for 20 min as their active esters of hydroxybenzotriazole in *N*-methylpyrrolidone in 4-fold excess.

The peptide resin (≈2 g) was treated with stirring for 2.5 h at 25 °C with a mixture of TFA (trifluoroacetic acid)/water/thioanisole/ethanedithiol (88: 5: 5: 2, by vol.) in the presence of phenol (2.2 g). The crude peptide was precipitated and washed by cold diethyl ether after filtration of the mixture, then centrifuged at 3000 g for 10 min, and the supernatant was discarded. The crude peptide was dissolved in water and then lyophilized.

The peptide was then solubilized, at 1 mM peptide concentration, in 0.2 M Tris/HCl buffer, pH 8.3, and stirred in the air to allow oxidative refolding (48 h, 25 °C). The folded and oxidized crude Bot33 peptide was purified by reverse phase HPLC column (Perkin-Elmer; C18 Aquapore ODS 20 m, 250 mm × 10 mm) with a 60 min linear gradient of 0–35% buffer B (0.08% (*v/v*) TFA in acetonitrile) in buffer A (0.1% (*v/v*) TFA in water) with a flow rate of 1 mL/min. The detection was performed at 230 nm.

The identity and homogeneity of the synthetic Bot33 peptide were evaluated by MALDI-TOF (laser-assisted matrix-desorption ionization-time-of-flight) MS molecular weight determination, N-terminal amino acids sequence analysis for the 5 first amino acids, and by Edman degradation of an analytical C18 reverse phase HPLC co-elution with the native peptide.

### 4.7. Electrophysiology

For the expression of Kv channels (mammalian rKv1.1, rKv1.2, hKv1.3, rKv1.4, rKv1.5, rKv1.6, and Drosophila *Shaker* IR) in *Xenopus laevis* oocytes, the linearized plasmids were transcribed using the T7 or SP6 mMESSAGE-mMACHINE transcription kit (Ambion, Carlsbad, CA, USA). The harvesting of stage V–VI oocytes from anesthetized female *Xenopus laevis* frog was previously described [37] in compliance with the regulations of the European Union (EU) concerning the welfare of laboratory animals as declared in Directive 2010/63/EU. The use of *X. laevis* oocytes was approved by the Animal Ethics Committee of the KU Leuven with the license number P186/2019. Oocytes were injected with 50 nl of cRNA at a concentration of 1 ng/nl using a microinjector (Drummond Scientific, Broomall, PA, USA). The oocytes were incubated in a solution containing 96-mM NaCl, 2-mM KCl, 1.8-mM CaCl_2_, 2-mM MgCl_2_, and 5-mM HEPES (pH 7.4), supplemented with 50 mg/l gentamycin sulfate.

Two-electrode voltage-clamp recordings were performed at room temperature (18–22 °C) using a Geneclamp 500 amplifier (Molecular Devices, Downingtown, PA, USA) controlled by a pClamp data acquisition system (Axon Instruments, Union City, CA, USA). Whole-cell currents from oocytes were recorded 1–4 days after injection. Bath solution composition was 96-mM NaCl, 2-mM KCl, 1.8-mM CaCl_2_, 2-mM MgCl_2_, and 5-mM HEPES (pH 7.4). Toxins were applied directly to the bath. Resistances of both electrodes were kept between 0.8 and 1.5 MΩ. The elicited currents were filtered at 0.5 kHz and sampled at 2 kHz using a four-pole low-pass Bessel filter. Leak subtraction was performed using an -P/4 protocol. Only data obtained from cells exhibiting currents with peak amplitude below 2 μA were considered for analysis. Kv1.1–Kv1.6 and Shaker currents were evoked by 250-ms depolarizations to 0 mV followed by a 250-ms pulse to −50 mV from a holding potential of −90 mV. All data were analyzed using pClamp Clampfit 10.1 (Molecular Devices, Downingtown, PA, USA) and Origin 7.5 software (Originlab, Northampton, MA, USA).

### 4.8. In Silico Analysis

#### 4.8.1. Molecular Modeling of Bot33 and Kv1.3 Potassium Channel

The three-dimensional structures (3D) of the Bot33 toxin and Kv1.3 potassium channel were generated by homology modeling with the program Modeller 9.24 [38]. The sequence of the voltage-gated potassium channel subunit Kv1.3 was extracted from UniProt under the accession number P22001 (https://www.uniprot.org/uniprot/P22001 (25 November 2008)). Its amino acid sequence was compared with other sequences retrieved from the NCBI database using FASTA and BLAST. Using only the pore domain of the channel crystal structure with MMTSB tools, the structure of the Kv1.3 potassium channel was predicted.

Bot33 homologous toxins, with known 3D structure, were identified by a Blast2 [39] search of the PDB database (RCSB organization) using the sequence of Bot33 as an entry. Multiple sequence alignment (MSA) is based on percent identity with the first member of each alpha-KTx subfamily. We have parameterized our research by selecting only the peptides derived from scorpion venom and the sequences which represent the best hits of BLAST have been selected.

After visualization, the structures of BMKTX toxin from the scorpion venom of *Buthus martensi Karsch* solved by NMR (PDB code 1BKT) [40], and the crystal structure of the kv1.3 (PDB code 7EJ1) [41] were used as templates. For each model, 1000 conformers were generated, and their DOPE scores were calculated. The best model of the Kv1.3 potassium channel and Bot33 toxin that present the lowest DOPE values were retained [42]. All structures were validated using Ramachandran plot analysis (PROCHECK), ProsaII, and Verify 3D. All structures were visually explored using a PyMOL molecular Viewer [43]. The structural models of Bot33_mM1 (Leu/Lys27), Bot33-m2 (Leu/Met27), and Bot33_mM32 (Leu/Lys27 and Asn/Tyr) were generated by simply mutating the 27th (Leu to Lys), (Leu to Met), and 36th (Asn to Tyr) and the 27th (Leu to Lys) amino acids, respectively using the same template solved by NMR (PDB code 1BKT).

#### 4.8.2. Toxin–Channel Docking Study

Bot33, Bot33/M27, and Bot33/K27_Y36 were analyzed to identify their interactions with the Kv1.3 potassium channel KCNA3_HUMAN from *Homo sapiens* (UniProtKB ID P22001: https://www.uniprot.org/uniprot/P22001 (25 November 2008). All the 3D structures were built through docking analysis using the ClusPro protein–protein docking software [44]. One thousand lowest energy structures were generated to find ten models defined by centers of highly populated clusters. For docking analysis, the retained complex presents the lowest energy compared to the other structures.

## Figures and Tables

**Figure 1 molecules-27-07278-f001:**
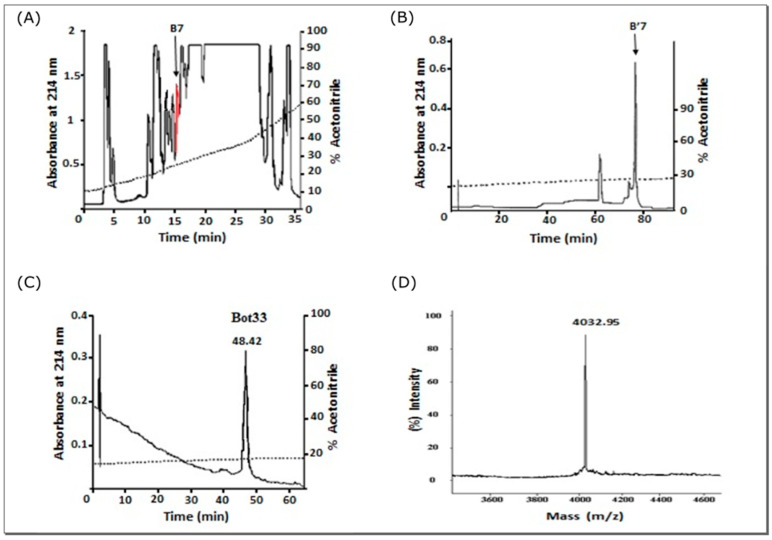
Purification of Bot33 from *Buthus occitanus tunetanus* scorpion venom. (**A**) Chromatography of fraction BotG50 on C18-RP-HPLC column. (**B**) Chromatography of fractionB7′on C18-RP-HPLC, B’7 was collected. (**C**) Bot33 purified from the fraction B’7 on C18-RP-HPLC. It is collected at 48.04 min. (**D**) Mass spectrometry of native Bot33 using MALDI-TOF.

**Figure 2 molecules-27-07278-f002:**
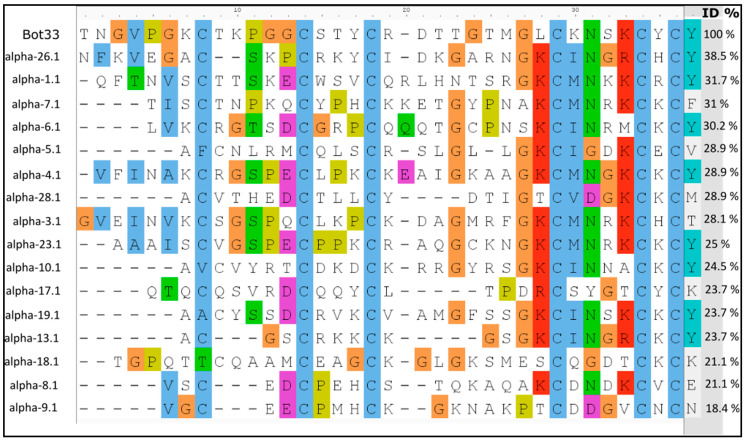
Sequence alignment. Alignment of Bot33 with the first members of each alpha-KTx subfamily described up to date. Cysteine residues are highlighted in red. The numbers next to each toxin’s name represent the percentage of identity (ID) with Bot33.

**Figure 3 molecules-27-07278-f003:**
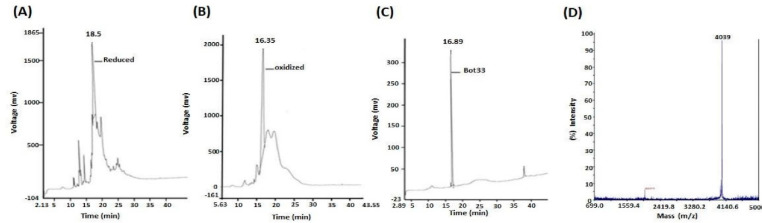
Bot33 at different stages of chemical synthesis: Analytical profiles of C18 reverse phase HPLC of Bot33 at different stages of its chemical composition synthesis. (**A**) The crude reduced peptide after the final TFA treatment. (**B**) The crude peptide after 48 h of oxidative folding. (**C**) Folded/oxidized purified Bot33. (**D**) The purity and identity of Bot33 assessed by mass spectrometry.

**Figure 4 molecules-27-07278-f004:**
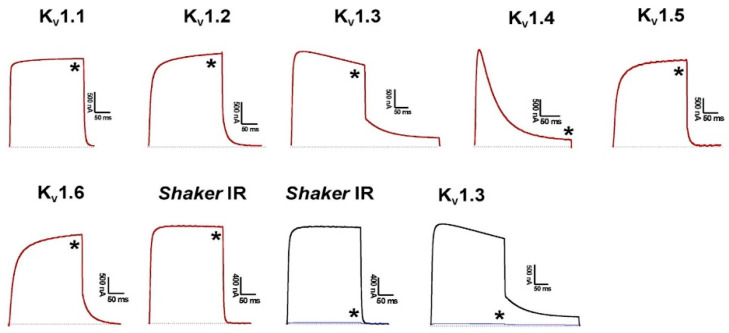
Differential effects of 1 µM toxin on Kv isoforms expressed in *Xenopus laevis* oocytes. The dotted line indicates the zero-current level. The black line indicates the current in control conditions, the red line indicates the steady-state current in the presence of 1 µM Bot33, the blue line indicates the steady-state current in the presence of 1 µM ChTx, while the * indicates the selected trace after application of toxin.

**Figure 5 molecules-27-07278-f005:**
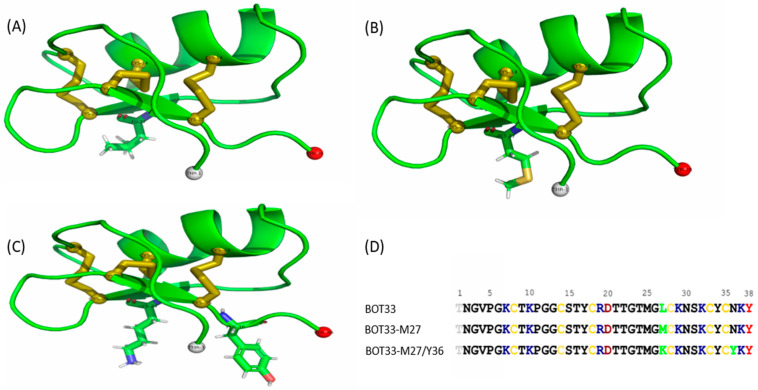
Three-dimensional structure modeling of the Bot33 toxin and its mutated peptides. (**A**) Three-dimensional model of the Bot33 toxin generated by homology modeling. (**B**) Bot33/M27. (**C**) Bot33/K27_Y36. (**D**) Multiple sequence alignment of Bot33 and its sequence mutation.

**Figure 6 molecules-27-07278-f006:**
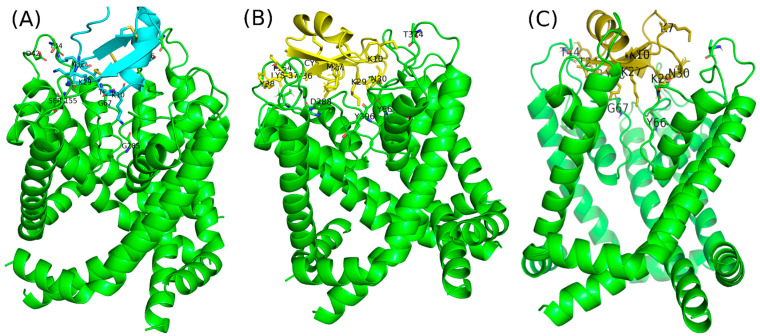
Protein–Protein docking of the Bot33 toxin and its mutations with the KV1.3 potassium channel. (**A**) Docking complex of Bot33 and Kv1.3 channel. (**B**) Docking complex Bot33/M27–Kv1.3 potassium channel reveals that the majority of basic residues are implicated in the interaction. (**C**) The interaction of Bot33/K27_Y36 with KV1.3 shows that the beta-sheet of the toxin interacts with the selectivity filter residues of the channel. The pore is physically blocked with the K27 lateral chain of Bot33/K27_Y36.

## Data Availability

Data are available on request from the authors.

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
