# Peer review of "Purification and Characterization of Bot33: A Non-Toxic Peptide from the Venom of Buthus occitanus tunetanus Scorpion"

_molecules, 2022, doi:10.3390/molecules27217278_

Round 1
Reviewer 1 Report
NOTE: In the previous and/or following lines, words that should be in italics are not italized, and characters that should be in superscript or subscript are not.
This manuscript reports on the purification and chemical and electrophysiological characterization of a novel peptide, Bot33, from the venom of the scorpion Buthus occitanus tunetanus.
Bot33 has less than 40% similarity with other known alpha-KTx families and the authors suggest that it belongs to a new family of scorpion toxins, and that it is the first member of the alpha-KTx33 family.
Natural peptide Bot33 did not display activity on mouse and neither did the synthetic peptide on Xenopus laevis oocytes expressing seven subtypes of potassium channels.
Therefore, the authors performed in silico molecular modelling and docking experiments of native and mutated Bot33 and they found explanations for the absence on biological effects and a probable different mode of action of peptide Bot33.
Thus, this study has contributed to the knowledge of the diversity of scorpion toxins and their modes of action.
POINTS:
Lines 115, 116, and 125: Please, review the nomenclature used for the novel toxin; in the Abstract it appears as Bot33, as it appears in the Title, the last paragraph of the Introduction, etc.
In the Discussion, more details might be provided on the rationale to assign peptide Bot33 to the KTx33 subfamily.
MINOR ISSUES:
Line 5: Please, change 4 to 3; please, change 3 to 4; please, change 3,* to 4,*
Line 14: Please, change 3 to 4 and move after new 3
Line 16: Please, change 4 to 3, and move after 2
Line 20: Please, change "Bot33" to "Bot33 is"
Line 23: Please, change "and the first" to "and it is the first"
Line 28: Please, apply italics to "Xenopus laevis"; please, change ", It was found" to ", and it was found"
Line 29: Please, change "on Kv" to "on seven Kv"
Line 49: Please, change "origin" in order not to repeat this word, used in the previous line
Lines 57-59: Strictly "its small size and strong cross-linking," are not the only determinants for "specific blocking activities"; the particular 3D structure of each member play a critical role. Please, rephrase.
Line 61: Please, omit the comma
Line 77: Please, check the current name of this company
Line 81: Please, change "and allowed" to "and it allowed"
Line 87: The general quality of the chromatograms should be improved and the names of the axis should be in English
Line 93: Please, change "4033.95" to "4032.95", because the mass of H+ must be subtracted from the m/z value (4033.95); please, change "almost identical" to "very close"
Line 94: Please, change "4032" to "4035.62"
Line 96: Please, change "motif Compared" to "motif. Compared"
Line 99: Please, check these values; "38.5" do not appear twice in Figure 2
Lines 97 and 99: Please, check whether you are referring to similarity (twice in this paragraph) or to identity; it should be pointed out that in the legend to Figure 2 "percent of identity" is used
Line 108: Please, remove unnecessary spaces between "with" and "the"
Line 109: Please, remove unnecessary spaces between "subfamily" and "described"
Line 110: Please, change "Id" to "ID", as used in the table itself
Figure 2: This reviewer considers that other identities between/among 2 or more sequences should be highlighted with other colors, in order to point out to resemblances between/among other toxins
Line 111: Please, omit unnecessary upper-case letters in the headings; 2.1, 2.2, and 2.3 do not contain them
Lines 112-114: Please, include a space between numbers and units
Line 127: Please, change to (M + H)+
Line 128: Please, change "value derived" to "value (4041.62) derived"
Line 138: Please, omit unnecessary upper-case letters in the headigs; 2.1, 2.2, and 2.3 do not contain them
Lines 140 and 141: Please, omit unnecessary upper-case letters in the headigs; 2.1, 2.2, and 2.3 do not contain them
Line 144: Please, change show on"" to "show any effect on"
Line 162: Please, put "Figure 5C" between parentheses
Line 167: Please, change "bridge" to "bridges:"
Figure 5: The alignment in panel D is incomplete; please, correct it
Line 181: Please, for uniformity, change "Met 27" to "M27"
Line 197: Please, change "mutation" to "mutations"
Line 205: Please, omit "any"
Line 208: Please, change "and" to "to"
Line 213: Please, change "and" to "to"
Lines 214-215: Please, change "of the channel were" to "of the channel, respectively, were"
Line 219: Please, omit the comma
Line 226: Please, change "," to "and"
Line 228: Please, change "," to "and"
Line 233: Please, change "(Figure 2) suggesting" to "(Figure 2), suggesting"
Line 243: Please, omit the comma
Line 255: Please, change "sheet, in silico mutagenesis studies" to "sheet. During the in silico mutagenesis studies"
Line 260: Please, change "toxins i.e." to "toxins, i.e."; please, change "toxins i.e. Lys-27 " to "toxins, i.e. Lys-27 "
Line 261: Please, change "Tyr38" to "Y38"
Line 262: Please, change "D288, therefore, M27" to "D288; therefore, M27"
Line 267: Please, omit "the"
Line 268: Please, omit "complex"
Line 271: Please, change "Bot33/Tyr36-Lys27" to "Tyr36 and Lys27"
Lines 282, 284, and 287: Please, include a space between value and unit
Line 284: Please, check the current name of this company and include details as done in line 302
Line 296: Please, omit the comma
Lines 302 and 305: "3" and "2" in "CH3CN/H2O" should be in subscript
Lines 303 and 305: Please, change "1-10 pmol. μl-1" to "1-10 pmol/μl"
Line 306: Please, change "of" to "aliquot of"
Line 315: The use of "Fmoc-amide resin" implies that the natural toxin is amidated, but nothing is said about the status of the C-terminus of this toxin in Results. Please, in Results, explain this and justify this assumption, because not all alpha-KTx have amidated C-termini.
Line 318: Please, verify this supplier (SDS) and include the full name and details
Line 329: Please, change ";" to "<space>during"
Line 333: Please, change "bot33" to "Bot33"
Line 334: Please, include a space between values and units (twice)
Line 339: Please, omit "amino acid analysis of" because no data are reported on this matter; otherwise, include the data on amino acid analysis in Results
Line 334: "oocytes should not be in italics; please, correct"
Line 349: "X. laevis" should be in italics
Line 366: Please, provide the additional details, given for other suppliers
Line 369: "V" in "KV1.3" should be in lowercase
Line 380: Please, omit "were"
Line 383: Please, change "toxin present" to "toxin that present"
Lines 394-395: Please, change "[40] 1000 lowest" to "[40]. One thousand lowest"
Line 400: Please, change "Supervised" to "supervised"
Line 401: Please, change "S.P. pharmacology" to "S.P. performed the pharmacology"
Line 412: The "T" of "The" should not be in boldface
References: Please, assure that all references comform to the current styole required by the journal; for example, some scientific names are not in italics
Author Response
Reviewer #1.
NOTE: In the previous and/or following lines, words that should be in italics are not italized, and characters that should be in superscript or subscript are not.
This manuscript reports on the purification and chemical and electrophysiological characterization of a novel peptide, Bot33, from the venom of the scorpion Buthus occitanus tunetanus.
Bot33 has less than 40% similarity with other known alpha-KTx families and the authors suggest that it belongs to a new family of scorpion toxins, and that it is the first member of the alpha-KTx33 family.
Natural peptide Bot33 did not display activity on mouse and neither did the synthetic peptide on Xenopus laevis oocytes expressing seven subtypes of potassium channels.
Therefore, the authors performed in silico molecular modelling and docking experiments of native and mutated Bot33 and they found explanations for the absence on biological effects and a probable different mode of action of peptide Bot33.
Thus, this study has contributed to the knowledge of the diversity of scorpion toxins and their modes of action.
Authors thank the reviewer #1 for the positive comments.
POINTS:
> Lines 115, 116, and 125: Please, review the nomenclature used for the novel toxin; in the Abstract it appears as Bot33, as it appears in the Title, the last paragraph of the Introduction, etc.
We revised the text to address your concerns; we updated the Title, the Abstract, and the last paragraph of the Introduction.
>In the Discussion, more details might be provided on the rationale to assign peptide Bot33 to the KTx33 subfamily.
We agree with the reviewer that the lack of function precludes classifying Bot33 as a KTx family member. Therefore, we have now rewritten the text, annotating Bot33 as a venom peptide and hope that it is now more clear.
MINOR ISSUES:
Line 5: Please, change 4 to 3; please, change 3 to 4; please, change 3,* to 4,*: done
Line 14: Please, change 3 to 4 and move after new 3: done
Line 16: Please, change 4 to 3, and move after 2: done
Line 20: Please, change "Bot33" to "Bot33 is»: done
Line 23: Please, change "and the first" to "and it is the first": done
Line 28: Please, apply italics to "Xenopus laevis"; please, change ", It was found" to ", and it was found": done
Line 29: Please, change "on Kv" to "on seven Kv": done
Line 49: Please, change "origin" in order not to repeat this word, used in the previous line: done
>Lines 57-59: Strictly "its small size and strong cross-linking," are not the only determinants for "specific blocking activities"; the particular 3D structure of each member plays a critical role. Please, rephrase.
As suggested by the reviewer, this is now rephrased.
Line 61: Please, omit the comma: done
Line 77: Please, check the current name of this company: done
Line 81: Please, change "and allowed" to "and it allowed": done
> Line 87: The general quality of the chromatograms should be improved and the names of the axis should be in English
We have improved the quality of the chromatograms and the names of the axes were written in English. We hope to find them well presented now.
>Line 93: Please, change "4033.95" to "4032.95", because the mass of H+ must be subtracted from the m/z value (4033.95); please, change "almost identical" to "very close"
As suggested by the reviewer, this is now fixed.
Line 94: Please, change "4032" to "4035.62": done
Line 96: Please, change "motif Compared" to "motif. Compared": done
Line 99: Please, check these values; "38.5" do not appear twice in Figure 2
As suggested by the reviewer, this is now fixed.
> Lines 97 and 99: Please, check whether you are referring to similarity (twice in this paragraph) or to identity; it should be pointed out that in the legend to Figure 2 "percent of identity" is used.
We totally agree with the comment made by the reviewer, this is now rephrased and expressed as identity percentage.
Line 108: Please, remove unnecessary spaces between "with" and "the": done
Line 109: Please, remove unnecessary spaces between "subfamily" and "described": done
Line 110: Please, change "Id" to "ID", as used in the table itself: done
> Figure 2: This reviewer considers that other identities between/among 2 or more sequences should be highlighted with other colors, in order to point out to resemblances between/among other toxins
We thank the reviewer for this remark. A new figure 2 was added in the revised manuscript highlighting similarities between different toxins with different colors.
Line 111: Please, omit unnecessary upper-case letters in the headings; 2.1, 2.2, and 2.3 do not contain them
As suggested by the reviewer, this is now fixed.
Lines 112-114: Please, include a space between numbers and units
As suggested by the reviewer, this is now fixed.
Line 127: Please, change to (M + H)+: done
Line 128: Please, change "value derived" to "value (4041.62) derived": done
Line 138: Please, omit unnecessary upper-case letters in the headigs; 2.1, 2.2, and 2.3 do not contain them: done
Lines 140 and 141: Please, omit unnecessary upper-case letters in the headigs; 2.1, 2.2, and 2.3 do not contain them: done
Line 144: Please, change show on"" to "show any effect on": done
Line 162: Please, put "Figure 5C" between parentheses: done
Line 167: Please, change "bridge" to "bridges:": done
Figure 5: The alignment in panel D is incomplete; please, correct it: done
Line 181: Please, for uniformity, change "Met 27" to "M27": done
Line 197: Please, change "mutation" to "mutations": done
Line 205: Please, omit "any": done
Line 208: Please, change "and" to "to": done
Line 213: Please, change "and" to "to": done
Lines 214-215: Please, change "of the channel were" to "of the channel, respectively, were": done
Line 219: Please, omit the comma: done
Line 226: Please, change "," to "and": done
Line 228: Please, change "," to "and": done
Line 233: Please, change "(Figure 2) suggesting" to "(Figure 2), suggesting": done
Line 243: Please, omit the comma: done
Line 255: Please, change "sheet, in silico mutagenesis studies" to "sheet. During the in silico mutagenesis studies": done
Line 260: Please, change "toxins i.e." to "toxins, i.e."; please, change "toxins i.e. Lys-27 " to "toxins, i.e. Lys-27 ": done
Line 261: Please, change "Tyr38" to "Y38": done
Line 262: Please, change "D288, therefore, M27" to "D288; therefore, M27": done
Line 267: Please, omit "the": done
Line 268: Please, omit "complex": done
Line 271: Please, change "Bot33/Tyr36-Lys27" to "Tyr36 and Lys27": done
Lines 282, 284, and 287: Please, include a space between value and unit:
Line 284: Please, check the current name of this company and include details as done in line 302
The current name of this company is “Beckman Coulter”.
Line 296: Please, omit the comma: done
Lines 302 and 305: "3" and "2" in "CH3CN/H2O" should be in subscript: done
Lines 303 and 305: Please, change "1-10 pmol. μl-1" to "1-10 pmol/μl": done
Line 306: Please, change "of" to "aliquot of": done
>Line 315: The use of "Fmoc-amide resin" implies that the natural toxin is amidated, but nothing is said about the status of the C-terminus of this toxin in Results. Please, in Results, explain this and justify this assumption, because not all alpha-KTx have amidated C-termini.
We apologize for this mistake. The used resin is not amidated, because the C-term of our peptide is undoubtedly carboxylated since de theoretical mass of the monoisotopic peptide is of 4032.754, as calculated by the peptide Mass Calculator program available on: https://www.peptidesynthetics.co.uk/tools/.
This mistake is now corrected
>Line 318: Please, verify this supplier (SDS) and include the full name and details
S.D.S (Solvents Documentation Synthesis) Address:quart Valdonne Postal code:13124 Town/Department:Peypin (Bouches-du-Rhône).
Line 329: Please, change ";" to "<space>during":done
Line 333: Please, change "bot33" to "Bot33": done
Line 334: Please, include a space between values and units (twice): done
> Line 339: Please, omit "amino acid analysis of" because no data are reported on this matter; otherwise, include the data on amino acid analysis in Results:
It was a mistake. We did the N-terminal sequencing for the 5 first amino acids, by Edman degradation.
Line 334: "oocytes should not be in italics; please, correct": done
Line 349: "X. laevis" should be in italics: done
Line 366: Please, provide the additional details, given for other suppliers: done
Line 369: "V" in "KV1.3" should be in lowercase: done
Line 380: Please, omit "were": done
Line 383: Please, change "toxin present" to "toxin that present": done
Lines 394-395: Please, change "[40] 1000 lowest" to "[40]. One thousand lowest": done
Line 400: Please, change "Supervised" to "supervised": done
Line 401: Please, change "S.P. pharmacology" to "S.P. performed the pharmacology": done
Line 412: The "T" of "The" should not be in boldface: done
>References: Please, assure that all references comform to the current style required by the journal; for example, some scientific names are not in italics
The authors thank the reviewer for pointing out this observation. All the references are reviewed and addressed according to the journal requirement.
Reviewer 2 Report
The authors in this study identify a toxin namely Bot33 from the venom of Buthus occitanus. The new toxin is novel in a sense as it has Leucine amino acid in position number 27 equivalent to lysine. But apparently the toxin is inactive on all Kv channel subtypes upon electrophysiological screening, and no other channel subtypes have been tested to show its activity.
Even though the authors mentioned that each member of alpha KTX family exhibits blocking of either Kv channels and/or KCa channels. However in this study only testing was reported on Kv,s. I would like to see the activity of Bot 33 on KCa channels as well as if to see if its active there. There wasnt any explanation given for this in discussion as well.
Section 2.4 suggests no toxicity seen with Bot33 upon ICV injection to mice. But the original results are not reported in the manuscript in the form of a timescale graph. What was the effect of Bot33 on analgesia? Has any analgesic activity been tested in mice with Bot33? The results needed to be more explanatory.
The legends in Figure 4 are misleading. It mentions "the blue line indicates the steady-state current in the presence of 1 μM Bot33 while the red line 152 indicates the steady-state current in the presence of 1 μM ChTx" But the original traces colours are otherway around.
Author Response
Reviewer #2.
Comments and Suggestions for Authors
The authors in this study identify a toxin namely Bot33 from the venom of Buthus occitanus. The new toxin is novel in a sense as it has Leucine amino acid in position number 27 equivalent to lysine. But apparently the toxin is inactive on all Kv channel subtypes upon electrophysiological screening, and no other channel subtypes have been tested to show its activity.
We would like to thank the reviewer#2 for this comment. We agree with the reviewer and have now changed the nomenclature of the peptide and we now describe it as a venom peptide. This was corrected throughout the text of the manuscript and particularly in the title.
>Even though the authors mentioned that each member of alpha KTX family exhibits blocking of either Kv channels and/or KCa channels. However in this study only testing was reported on Kv,s. I would like to see the activity of Bot 33 on KCa channels as well as if to see if its active there. There wasn't any explanation given for this in discussion as well.
We thank the reviewer for this valuable remark. We agree with the reviewer that it is interesting to investigate the activity of Bot33 on KCa channels. Unfortunately, at present we do not have enough material of Bot33 to test on KCa channels. Furthermore, it should be noted that KCa channels are difficult to express in Xenopus laevis oocytes which is the reason that these channels were not included in the present study. We intend to check the activity of Bot33 on KCa channels in future studies. However, we do feel that these experiments are beyond the scope of this work which is the investigation of Bot33 on Kv channels. We sincerely hope that the reviewer can agree with our reasoning.
> Section 2.4 suggests no toxicity seen with Bot33 upon ICV injection to mice. But the original results are not reported in the manuscript in the form of a timescale graph.
A table reporting a timescale toxicity test was added in the supplementary data (Table 1).
> What was the effect of Bot33 on analgesia? Has any analgesic activity been tested in mice with Bot33? The results needed to be more explanatory.
To date no short scorpion toxin (less than 6KDa) has been shown to have analgesic activity. On the other hand, scorpion peptides with an analgesic effect are sodium channel modulators, which is not the case of Bot33. On the other hand, by testing the toxicity we didn't see any effect on the behavior of the injected mice. Therefore testing the analgesic effect of Bot33 is not justified, especially when we can’t afford it.
>The legends in Figure 4 are misleading. It mentions "the blue line indicates the steady-state current in the presence of 1 μM Bot33 while the red line 152 indicates the steady-state current in the presence of 1 μM ChTx" But the original trace colors are the other way around.
This has been corrected.
Reviewer 3 Report
General opinion.
This is an interesting article showing that a lot could be scientifically made (excellent methodology, excellent approach, excellent in-silico modeling, excellent synthesis and activity tests) without exiting result. I would have recommend this paper as a method paper instead of a “discovery paper”.
In the best case, this paper confirms the need to have a Lys or Arg in position 27. This is not really an innovation. Is it?
Here are some comments to be consider by authors.
1. Introduction does not clearly state the goal of the study.
2. The choice of this species is not explained nor justified. Are there previous toxins identified from this species or similar once that make expect to find more ?
3. Please add a figure of Buthus occitanus.
4. Line 48 : What are « multimeric proteins « related to ? The previous paragraph was about toxins. Is it the same subject or not. Clarify this. The reference to human pathologies is more than confusing.
5. Line 51. Make a new paragraph begining at « Alpha-Ktx »
6. Line 59. Make a new paragraph begining at « Here we describe »
7. Line 66. The isolation, identification and characterization are not done « in the paper » maybe « in the study that this paper is the report ».
8. Line 80. Sentence without subject. Add one please.
9. Line 95 and succ : Seqeunce analysis.
>Bot33
TNGVPGKCTKPGGCSTYCRDTTGTMGLCKNSKCYCNNKY
- A FASTA or even text representation of the obtained amino acid sequence must be presented in the text and not only in the figure. Impossible to get it from the text. Please provide one. Or at least for reviewers.
- The deposition of the sequence on public repository (genBank) should be made and the accession number be available before publication. Or at least for reviewer.
- How was the sequences used for the MSA Chosen? Is the MSA presented in the article made with best BLAST hits ? This is unclear in the text.
- With a similarity of less than 40 %, from what is presented, it appears that very few position except from the cysteine framework is conserved. My question is : Is it possible that this cysteine framework belong to something else ? Could this peptide be a fragment of something else? Is this specific framework unique to the proposed homologous (in this case alpha-Ktx) ? Since authors seems to search for alpha-Ktx, is it possible that they only focused on their targeted family.
- A very raw BLAST on Uniprot allowed to get homologous sequences with a L at the same position, suggesting bot33 may belong to an already existing family. See A0A2J6MEH6 or S8CIG1. Neither the words « kunitz » nor the term « knottin » are present in the paper. However a raw BLAST search with the sequence fished out sequences related to that keywords ;
We therefore recommend a deeper bioinformatic analysis.
10. Regarding author conclusion (line 233). Concluding that a peptide not exhibiting a blocking activity is a new ALPHA-neurotoxin represent a “contradictio in terminis”. Indeed, been an ALPHA suppose to be active. We therefore recommend to find another family name if necessary. At first glance, it seems that this peptide is not from a “new” family and that not even deserve to create a new family. However, not being myself expert in scorpion toxin family nomenclature, I propose to first attach this inactive peptide to Knottin-like Scorpion toxin and or kunitz-like scorpion toxin.
Author Response
Reviewer #3.
Review Report Form
Comments and Suggestions for Authors
General opinion.
This is an interesting article showing that a lot could be scientifically made (excellent methodology, excellent approach, excellent in-silico modeling, excellent synthesis and activity tests) without exiting results. I would have recommended this paper as a method paper instead of a “discovery paper”.
>In the best case, this paper confirms the need to have a Lys or Arg in position 27. This is not really an innovation. Is it?
In fact our work confirms the importance of the positive charged residue at position 27 and this was already shown by other authors [Park et al. 1992; Goldstein et al. 1994; Swartz et al. 2013; Zhu et al.2014]. However Naseem, M.U et al., 2022 recently reported that the Cm28 peptide inhibited voltage-gated K+ channels KV1.2 and KV1.3 with Kd values of 0.96 and 1.3 nM, respectively, despite lacking this residue. In our work we clearly showed that, besides Lys27 other residues, such as hydrophilic residues: T, K, N and hydrophobic residues M, Y, should be present in the toxin sequence to allow its interaction with K+ channel. This is now added in the conclusion: line 260.
>1. Introduction does not clearly state the goal of the study.
The goal of the study is now added to the actual version of the manuscript at the end of the introduction.
>2. The choice of this species is not explained nor justified. Are there previous toxins identified from this species or similar once that make us expect to find more ?
The Buthus occitanus tunetanus scorpion represents one of the most dangerous species. Only few peptides were reported from its venom [El Ayeb M et al.1983; Kharrat R et al 1997; Mejri T et al.2003; Mahjoubi-Boubaker B et al. 2004; ElFessi-Magouri R et al.2016 and 2015; Khamassi O et al.2018; Ben abderaazek R et al.2022].
This was highlighted in the revised version of the manuscript
>3. Please add a figure of Buthus occitanus.
The figure of scorpion Buthus occitanus tunetanus is added in the supplementary data (Figure S1).
>4. Line 48 : What are « multimeric proteins « related to ? The previous paragraph was about toxins. Is it the same subject or not. Clarify this. The reference to human pathologies is more than confusing.
We totally agree with the comment made by the reviewer, this is now rephrased.
>5. Line 51. Make a new paragraph begining at « Alpha-Ktx »
As suggested by the reviewer, this is now fixed.
>6. Line 59. Make a new paragraph begining at « Here we describe »
A new paragraph has been drafted.
>7. Line 66. The isolation, identification and characterization are not done « in the paper » maybe « in the study that this paper is the report ».
We agree with the reviewer and this was deleted from the manuscript.
>8. Line 80. Sentence without subject. Add one please.
We agree with the reviewer and this has been added.
- Line 95 and succ : Sequence analysis.
- A FASTA or even text representation of the obtained amino acid sequence must be presented in the text and not only in the figure. Impossible to get it from the text. Please provide one. Or at least for reviewers.
We thank the reviewer for this comment. This is now added in the actual version.
>Bot33
TNGVPGKCTKPGGCSTYCRDTTGTMGLCKNSKCYCNKY
- The deposition of the sequence on public repository (genBank) should be made and the accession number be available before publication. Or at least for reviewer.
We thank the reviewer for this comment. We have deposited the Bot33 sequence in the uniprot database “Section SPIN”, SPIN ID number SPIN200024302
- How was the sequences used for the MSA Chosen? Is the MSA presented in the article made with best BLAST hits ? This is unclear in the text.
We have modified the paragraph to improve clarity.
Multiple sequence alignment (MSA) is based on percent identity with the first member of each alpha-ktx subfamily. We have parameterized our research by selecting only the peptides derived from scorpion venom and the sequences which represent the best hits of BLAST have been selected.
I added a sentence in the Materials and methods section.
- With a similarity of less than 40 %, from what is presented, it appears that very few position except from the cysteine framework is conserved. My question is : Is it possible that this cysteine framework belong to something else ? Could this peptide be a fragment of something else? Is this specific framework unique to the proposed homologous (in this case alpha-Ktx) ? Since authors seems to search for alpha-Ktx, is it possible that they only focused on their targeted family.
The framework found in Bot33 is conserved for scorpion venom peptides that act on Kv channels or defensins. The cysteine stabilized alpha beta structure is a conserved structure found in many organisms. Since we do not have transcriptomic data, we do not know if Bot33 is a fragment of something else. However, to the best of our knowledge, bigger proteins consisting in part of a peptide with such a cysteine framework have never been described before in scorpion venom. We agree with the reviewer that the activity of Bot33 might be different from Kv channel blockage. Since we could not prove that Bot33 acts on Kv channels, we now denote Bot33 as a venom peptide and no longer as an alpha-KTx.
- A very raw BLAST on Uniprot allowed us to get homologous sequences with a L at the same position, suggesting bot33 may belong to an already existing family. See A0A2J6MEH6 or S8CIG1. Neither the words « kunitz » nor the term « knottin » are present in the paper. However a raw BLAST search with the sequence fished out sequences related to that keywords ;
We therefore recommend a deeper bioinformatic analysis.
We agree with the reviewer, the search with (BLAST) on uniprot gives us the results mentioned by the reviewer. (https://www.uniprot.org/blast/uniprotkb/ncbiblast-R20220815-013016-0810-38051808-p2m/overview).
On the other hand, we have parameterized our research by selecting only the peptides derived from scorpion venom. We have modified the paragraph to improve clarity.
Multiple sequence alignment (MSA) is based on percent identity with the first member of each alpha-ktx subfamily. We have parameterized our research by selecting only the peptides derived from scorpion venom.
I added a sentence in the Materials and methods section.
>10. Regarding author conclusion (line 233). Concluding that a peptide not exhibiting a blocking activity is a new ALPHA-neurotoxin represents a “contradictio in terminis”. Indeed, been an ALPHA is supposed to be active. We therefore recommend finding another family name if necessary. At first glance, it seems that this peptide is not from a “new” family and that not even deserve to create a new family. However, not being myself an expert in scorpion toxin family nomenclature, I propose to first attach this inactive peptide to Knottin-like Scorpion toxin and or kunitz-like scorpion toxin.
We agree with the reviewer that the lack of function precludes the annotation of Bot33 as a KTx. Following the reviewer’s remark, we have now rewritten the text. Bot33 is now described as a new scorpion venom-derived peptide. However, the structure of this peptide could not be attached to neither knotting-like nor kunitz-like scorpion toxins. Indeed, Kunitz-like proteins are peptides formed by about 60 amino acids (SkarzyÅ„ski T, 1992). On the other hand, the predominant secondary structural element associated with the cysteine ​​knot toxins is a two- or three-stranded β-sheet that is intimately associated with the cystine knot, however, Bot33 clearly adopts the Csαβ motif, as demonstrated by the molecular model of its 3D structure as well as its amino acid sequence alignment with the similar scorpion toxins (Figure A, B).
Figure A: Comparison of Bot 33 and Dendrotoxin ( Kunitz like).
Figure B: Comparison of Bot33 and Conotoxin (Knottin like).
Bibliographic references
- Zhu, S.; Peigneur, S.; Gao, B.; Umetsu, Y.; Ohki, S.; Tytgat, J. Experimental conversion of a defensin into a neurotoxin: implications for origin of toxic function. Mol Biol Evol. 2014, 31,546-59.
- Swartz, K.J. The scorpion toxin and the potassium channel. Elife. 2013, 21.
- Park, C.S.; Miller, C. 1992. Interaction of charybdotoxin with permeant ions inside the pore of a K+ channel. Neuron. 1992, 9, 307–13.
- Goldstein, S.A.; Pheasant, D.J.; Miller, C. 1994. The charybdotoxin receptor of a Shaker K+ channel: peptide and channel residues mediating molecular recognition. Neuron. 1994, 12, 1377–88.
- Naseem, M.U.; Carcamo-Noriega, E.; Beltran-Vidal, J.; Borrego, J.; Szanto, T.G.; Zamudio, F.Z.; Delgado-Prudencio, G.; Possani, L.D.; and Panyi, G. Cm28, a scorpion toxin having a unique primary structure, inhibits KV1.2 and KV1.3 with high affinity. J. Gen. Physiol. 2022, 8, 154.
6. Skarzynski, T. Crystal structure of alpha-dendrotoxin from the green mamba venom and its comparison with the structure of bovine pancreatic trypsin inhibitor. J Mol Biol. 1992, 224(3), 671-83.
Round 2
Reviewer 2 Report
The authors have edited the manuscript very well. I have no further suggestions and accept the manuscript for publication
Reviewer 3 Report
We thank authors for their effort that add more clarity in this paper. The presentation is really improved. We consider the draft ready for publication. The only concern remain the addition of the species picture in article and not in supplementary material.
As previously indicated, from our point of view, this paper remains a very good methodological paper than either an discovery nor original paper.
We let authors discuss this question with the Editor.